# Determinants of hypertension among diabetes patients attending selected comprehensive specialized hospitals of the Amhara Region, Ethiopia: An unmatched case-control study

**Makda Abate Belew** [1]*, **Teshager Woldegiorgis Abate**[2], **Alemshet Yirga Berhie**[2], **Eleni Dagnaw Abeje**[1], **Dawit Algaw Ayele**[3], **Melsew Dagne Abate**[4], **Rediet Akele Getu**[1], **Berihun Bantie**[5], **Sewnet Getaye Workie**[6]

1 Department of Nursing, School of Nursing and Midwifery, College of Medicine and Health Science, Debre Berhan University, Debre Berhan, Ethiopia, 2 Department of Adult Health Nursing, College of Medicine and Health Science, Bahir Dar University, Bahir Dar, Ethiopia, 3 Department of Nursing, Bahir Dar Health Science College, Bahir Dar, Ethiopia, 4 Department of Nursing, College of Health Sciences, Woldia University, Woldia, Ethiopia, 5 Department of Nursing, College of Medicine and Health Science, Debre Tabor University, Debre Tabor, Ethiopia, 6 Department of Public Health, School of Public Health, College of Medicine and Health Science, Debre Berhan University, Debre Berhan, Ethiopia

* sismakda@gmail.com

## Abstract

### Background

The coexistence of diabetes mellitus and hypertension is a worldwide public health problem causing significant morbidity, mortality, and decreased quality of life. Despite the increasing burden of hypertension among patients with DM, data on determinants of hypertension among patients with DM in the Amhara region of Ethiopia is scarce. Hence, this study identified determinants of hypertension among people with diabetes attending chronic disease follow-up clinics in the Amhara region of comprehensive specialized hospitals in Ethiopia.

### Method and materials

An institutional-based unmatched case-control study was conducted among 470 individuals with diabetes in the Amhara region's comprehensive specialized hospitals (Debre Berhan, Felege Hiwot, and Dessie Comprehensive specialized hospital). A multistage sampling technique was used to select participants for this study. We collected the data using standard questionnaires (short form of international physical activity questionnaire, Morisky medication adherence scale, patient health questionnaire, perceived dietary adherence scale, Oslo social support questionnaire, and alcohol use disorder identification test), physical measurements, and data extraction checklists. A multivariable binary logistic regression was fitted to identify determinants of hypertension, and we presented the findings using an adjusted odds ratio (AOR) with a 95% confidence interval (CI).

**Data Availability Statement:** All relevant data are within the paper and its Supporting Information files.

**Abbreviations:** 8-MMS, 8 Item Morisky Medication Adherence Scale; AOR, Adjusted Odds Ratio; AUDIT, Alcohol Use Disorders Identification Test; BMI, Body Mass Index; CI, Confidence Interval; COR, Crude Odd Ratio; CVDs, Cardiovascular Diseases; DM, Diabetes mellitus; IPAQ, : International Physical Activity Questionnaire; HC, : Hip Circumference; HTN, Hypertension; MET, Metabolic Equivalent Task; NCDs, Non-Communicable Diseases; OSLO, Oslo social support questionnaire; PDAQ, Perceived Dietary Adherence Questionnaire; PHQ 9, 9 Item Patient Health Questionnaire; WC, : Waist Circumference; WHR, Waist to Hip Ratio.

## Results

235 cases and 235 controls participated in this study. The median (IQR) age for the cases was 60 (52–66 = 14), and the mean age (± SD) for the controls was 51.72 (± 12.51). The significant determinants of hypertension with AOR [95% CI] were a lower level of physical activity: 1.82 [1.00, 3.31], depression: 2.00 [1.24, 3.21], family history of hypertension: 2.13 [1.34, 3.37], not having diabetic health education: 1.87 [1.18, 2.96], a longer duration of diabetes: 1.99 [1.05, 3.79], and poor glycemic control: 1.57 [1.01, 2.45].

## Conclusion

In this study, determinants that increase the risk of hypertension among people with diabetes mellitus were older age, physical inactivity, depression, family history of hypertension, not having diabetic health education, a longer duration of diabetes, and poor glycemic control.

## Introduction

Diabetes mellitus (DM) is a serious, chronic metabolic condition characterized by a state of hyperglycemia resulting from defects in insulin secretion, insulin action, or both. According to the 2021 estimation of the International Diabetes Federation (IDF), around 531 million adults aged 20 to 79 (10.5% of all adults in this age group) are living with diabetes. This figure is predicted to rise to 643 million by 2030 and 783 million by 2045. Among the forty-eight Sub-Saharan African countries and territories in the IDF Africa Region, Ethiopia has the fourth-largest number of people living with DM in 2021 [1]. According to a recent systematic review and meta-analysis, the pooled prevalence of DM in Ethiopia was 4.99% [2]. Globally, around 6.7 million adults (20–79) were estimated to die from diabetes or its complications, including hypertension, in 2021 [1].

Hypertension (HTN) among individuals with diabetes mellitus is a severe medical condition characterized by persistent elevation of systemic arterial blood pressure, defined as systolic blood pressure (SBP) of above 130 mmHg and diastolic blood pressure (DBP) of above 80 mmHg on two consecutive days or any prior diagnosis of HTN made by health personnel and taking antihypertensive drugs [3, 4]. The frequency of HTN among individuals with diabetes is twice that of individuals without diabetes and is reported in over two-thirds of people with type 2 diabetes [5]. The high incidence of hypertension among individuals with diabetes is attributed to different structural alterations secondary to endothelial dysfunction, insulin resistance in the nitric-oxide pathway, sodium fluid retention, the stimulatory effect of hyperinsulinemia on the sympathetic nervous activity, and the excitatory effect of hyperglycemia on renin-angiotensin-aldosterone system [1, 5–7].

The coexistence of DM with hypertension increases the risk of cardiovascular events by six-fold, doubles the risk of all-cause mortality and stroke, triples the risk of coronary heart diseases, and hastens the progression of both microvascular and macrovascular complications [8]. One study showed that the coexistence of DM with hypertension is attributed to the risk of death and cardiovascular events by 44% and 41%, respectively, compared with 7% (mortality risk) and 9% (cardiovascular risk) in people with diabetes alone [9]. Uncontrolled hypertension among individuals with diabetes predisposes to the development of heart attack, stroke, kidney disease, vision loss, sexual dysfunction, and peripheral arterial disease [5]. Moreover,

the development of HTN in individuals with diabetes complicates the treatment strategy, considerably impairs health-related quality of life (HRQoL), and causes a substantial economic impact [6, 10, 11].

Behavioural risk reduction and lifestyle modification (cessation of tobacco smoking, adhering to a healthy diet plan, engaging in physical activity, minimizing alcohol consumption, weight loss, a dietary approach to stop hypertension (DASH) style-based nutrition counseling, and reduced-sodium/salt intake) are strategies implemented in the past to control the development of hypertension among individuals with DM [12]. Despite implementing these strategies, many studies reported a high burden of hypertension among people with DM [6, 13–17]. So far, some determinants of hypertension in people with diabetes have been identified in some countries (United Arab Emirates, Jordan, Nigeria, Malaysia, Republic of Benin, Libya, Ethiopia), namely, advanced age, low educational status, unhealthy diet, family history of hypertension, poor glycemic control, and duration of diabetes [14–20]. The most common determinant was a longer duration of diabetes. We aimed to fill the existing research gap in our study area by examining multiple risk factors (depression, harmful alcohol consumption, poor social support, raised BMI, increased waist circumference, and raised waist-to-hip ratio) contributing to the development of hypertension in individuals with DM.

One of the IDF's suggested actions to lower the risk of CVD outcomes and chronic kidney disease among individuals with DM is to prevent the development of HTN. Although the need to reduce the burden of HTN among DM patients is studied extensively, how we reduce it in this population calls for further evidence. Despite the disproportionately high burden of HTN among individuals with DM, to the best of our knowledge, there are limited studies on determinants of Hypertension in Ethiopia, including our study area (Amhara region) [6, 11, 17, 21–23] conducted on this specific population. These studies have significant methodological limitations, including flawed participant population selection to explore determinant factors for developing HTN among DM patients. Therefore, this study addressed the issues mentioned above and identified determinants of HTN among DM patients that can be helpful for the effective prevention and control of the condition.

## Methods and materials

### Study setting, design, and period

An institutional-based unmatched case-control study was conducted from March 17, 2021, to April 18, 2021, among people with diabetes mellitus in the Amhara region's comprehensive specialized hospitals. There are eight comprehensive specialized hospitals in the area, with an estimated total number of individuals with DM reporting for follow-up care of approximately 18,573.

Three randomly selected comprehensive specialized (Debre-Berhan, Felege-Hiwot, and Dessie) hospitals were included. These hospitals' chronic disease follow-up clinics provide services on all working days and are approximately visited by twenty-five to thirty individuals with DM daily.

### Population

All adult individuals with diabetes mellitus attending chronic disease follow-up clinics in the Amhara region comprehensive specialized hospitals were the source population. Those with diabetes mellitus coexisting with hypertension and attending the chronic disease follow-up clinic for follow-up in the selected hospitals during the study period were considered cases, whereas those with diabetes and no hypertension served as controls.

Critically ill patients with DM or severe medical illness, newly diagnosed patients with diabetes who were not on antidiabetic medication within the last six months in a regular follow-up, and a history of hypertension at the time and before the diagnosis of DM were excluded from this study.

## Sample size determination and sampling procedure

Epi info version 7.2.3.1 was used to calculate the sample by assuming a 95% confidence interval (CI), 80% power, and a 1:1 case-to-control ratio. The Odds ratio (OR) and the proportion of potential predictor variables of hypertension among controls were taken from a study in Tigray, Ethiopia [23]. After considering adjustments for the expected non-response rate (10%) and design effect (1.5), the largest total sample size became 476.

A multistage sampling technique was used to select participants for this study. Initially, a list of all comprehensive specialized hospitals was taken from the Amhara health bureau. Of the eight comprehensive specialized hospitals in the Amhara region, three comprehensive specialized hospitals (Debre Berhan, Dessie, and Felegehiwot) were selected using simple random sampling. After random selection of the hospitals, the number of patients with diabetes with hypertension and without hypertension who took service from chronic follow-up clinics were taken from the chronic disease follow-up clinic registry book. Finally, we selected eligible cases for every two patients using systematic random sampling. For each case, one control that fulfils the inclusion criteria was selected for every seven patients using a systematic random sampling technique from chronic follow-up clinics of the same hospitals from which cases were drawn.

## Study variables

The dependent variable in the current study was hypertension among people with DM. Independent variables include sociodemographic, clinically related, psychosocial, and behavioural variables.

## Data collection tools and procedures

Data were collected using semi-structured interviewer-administered questionnaires, physical measurements, and data extraction checklists developed after reviewing different literature [6, 11, 16, 17, 19, 22–24]. Six BSC nurses collected the data and were supervised by three MSc nursing students. The first part of the questionnaire was sociodemographic data (age, sex, marital status, average family income, occupational status, and educational status).

**Psychosocial and behavioural variables.** Study participants' psychosocial and behavioural characteristics (level of physical activity, harmful alcohol consumption, diabetic medication adherence, adherence to a healthy diet, depression, and social support) were the second part of the questionnaire.

Participants' level of physical activity was assessed using the international physical activity questionnaire short-form (IPAQ-SF) [25]. This tool is designed to assess specific types of exercise, such as walking, moderate and vigorous-intensity activities done at work, as part of house and yard work, to get place to place, and in spare time for recreation, exercise, or sport. Study participants were asked to recall their activities of the last seven days preceding the interview. Data were reported as the metabolic equivalent of task minutes (MET-minutes per week) using the IPAQ screening protocol [25]. This tool has been used in studies conducted in Ethiopia [26, 27]. Participants were considered to have higher, moderate, or lower adherence to physical activity if they met any criterion for higher, moderate, or lower levels of physical activity following the IPAQ screening protocol [28].

The eight-item Morisky Medication Adherence Scale (MMAS-8) [29, 30] was used to measure the participants' Self-reported adherence to diabetic medication. It contains eight items, with binary scores (yes/no) for the first seven items and a 5-point Likert score for the last. The last item contributes a score between zero and one in 0.25-point increments on a 5-point scale assessing the frequency patients forget to take medications (never = 1, once in a while = 0.75, sometimes = 0.5, usually = 0.25, and all the time = 0). Each "no" response was rated "1," and each "yes" was rated "0" except for item 5 (reversed score), in which the response "yes" was rated "1" and "no" was rated "0" [31]. This tool was validated and used in a previous study conducted in Ethiopia [32, 33]. The total score is a summation of all MMAS-8 items. On MMAS-8, participants were considered to have good adherence to a medication when they scored 8, medium adherence to medication if they scored 6 to lower than 8, and low adherence to medication if they scored < 6 [30].

The perceived dietary adherence questionnaire (PDAQ), a nine-item tool, is also used for assessing dietary adherence [34]. The response is based on a seven-point Likert scale to answer the question, "On how many of the last 7 days did you. . ..?" Higher scores reflect higher adherence except for items 4 and 9, which reflect unhealthy choices (foods high in sugar or fat). For these items, higher scores reflect lower adherence. Therefore, for computing a total PDAQ score, the scores for these items were inverted. Participants were classified as having good adherence to a healthy diet if they ate a healthy diet for at least four days a week and as having poor adherence to a healthy diet if they had eaten a healthy diet for less than four days a week [34].

Patient Health Questionnaire 9 (PHQ-9), a nine-item self-report instrument (scoring ranges from 0 to 27), and a standardized, validated tool in East Africa, including Ethiopia, were used for assessing depression [35]. Participants with a score of five and above on the PHQ-9 were considered to have depression. On the PHQ-9 depression subscales, participants were sub-classified as experiencing no depression if the PHQ-9 score is 0–4, mild depression if the PHQ-9 score is 5–9, moderate depression if the PHQ-9 score is 10–14, moderately severe depression if the PHQ-9 score is 15–19, and severe depression if the PHQ-9 score is 20–27 [36].

Participants' level of social support was assessed using the 3-item Oslo social support scale (OSS-3) by asking them to rate the level of support they received from family and friends. This tool has been validated in various African countries [37] and used in studies conducted in Ethiopia [38]. On OSS-3, a participant with a score of 3–8 was considered as having "poor social support," 9–11 as having "moderate social support," and 12–14 as having "strong social support" [38].

The 10-item alcohol use disorder identification test (AUDIT) tool was used to assess alcohol consumption level (3 items), symptoms of alcohol dependence (3 items), and problems associated with alcohol use (4 items) [39, 40]. Participants who scored eight and above were considered hazardous alcohol consumers

Adherence to self-blood glucose monitoring: the patient was considered to adhere to blood glucose monitoring when they scored above the mean of the number of days [41].

**Clinically related variables.** Recent clinical-related and biochemical data like fasting blood glucose level, comorbidity, diabetes-related complications, duration of diabetes, and treatment modality were collected from the patient's records using data extraction checklists.

The participants were considered to have good glycemic control if the average of the last three fasting blood glucose levels was between 70mg/dL and 130mg/dL or poor glycemic control if the average of the last three fasting blood glucose levels was above 130 mg/dL [36].

**Anthropometric measurements.** After the participants stood with arms by the sides, feet positioned close together, and weight evenly distributed across the feet, waist circumference

(WC) was measured to the nearest 0.1 cm at the end of normal exhalation at the level of the iliac crest. WC was considered above average if the measurement was >94 cm for males and >80 cm for females [42].

The hip circumference of the patients was measured to the nearest centimetre at the largest maximum circumference of the buttocks. Both hip and waist circumferences were measured with stretch-resistant tape wrapping snugly around the participants. Waist to hip ratio (WHR) was calculated as WC (cm) divided by HC (cm) [42]. WHR was considered above average if the measurement was ≥ 0.90 for males and ≥ 0.85 cm for females [42].

The participants' heights were measured to the nearest centimetre using a stadiometer, with participants standing upright. Weight (in kilograms) was measured in light clothing using a calibrated Seca digital weighing scale (manufactured by Seca gmbh & co. kg/ model: 874, designed in Hamburg, Germany) and recorded to the nearest 0.1 kg. The scale was calibrated regularly, and the indicator was checked against zero reading before every measurement. Body mass index (BMI) was calculated as the weight divided by height squared (kg/m2). Participants were considered underweight if the BMI was less than 18.5 kg/m2, healthy weight if the BMI was 18.5 to 24.9 kg/m2, overweight if the BMI was 25 to 29.9 kg/m2, and obese if the BMI was 30kg/m2 or higher [42].

## Data processing and analysis

Data were checked for completeness, and each completed questionnaire was assigned a unique code. Epi Data version 3.1 and Statistical Package for Social Sciences [36] version 21 were used for data entry and analysis, respectively. The data were checked by visualizing, calculating frequencies, and sorting. After performing a normality test, continuous variables with normal distributions were reported as mean ± standard deviation (SD), while those with skewed distributions were reported as the median and interquartile range (IQR)(25th; 75th percentage), and categorical variables were summarized as frequency and percentage. The bivariate logistic regression model was fitted for each explanatory variable. Accordingly, we applied the statistical methodology for variable selection using P-value less than 0.2 in the bivariate analysis as a threshold to identify a candidate set of variables that will enter the multivariable model. Then, variables with a P-value of less than 0.05 in the multivariable analysis were considered statistically significant, and AOR with 95% CI was estimated to measure the strength of the associations. The model's fitness was checked using Hosmer and Lemeshow goodness-of-fit test statistics, giving a P-value of 0.463, suggesting that the data fit well with the model.

## Ethical consideration

Ethical clearance was obtained from the Institutional Review Board of Bahir Dar University, College of Medicine, and Health Sciences, with protocol number 079/2021. A formal approval letter for collecting data was submitted to the selected hospitals. Before data collection, informed consent was obtained from participants, and confidentiality of responses was maintained throughout the study.

## Results

### Sociodemographic characteristics of the study participants

A total of 470 individuals with diabetes (235 cases and 235 controls) were included in this study. Male participants comprised 51.5% of the controls and 50.2% of the cases. The mean (± SD) age for the controls was 51.72 (± 12.51), and the median age (IQR) for the cases was 60 (52–66 = 14). Occupation-wise, 25.1% of the cases were retired, and 24.3% of the controls were

**Table 1. Sociodemographic characteristics of study participants.**

| Variables | Cases (n = 235) | | Controls (n = 235) | |
|---|---|---|---|---|
| | Frequency | % | Frequency | % |
| Sex | | | | |
| Male | 118 | 50.2 | 121 | 51.5 |
| Female | 117 | 49.8 | 114 | 41.5 |
| Age (years) | | | | |
| ≤ 50 | 45 | 19.2 | 123 | 52.3 |
| 51–60 | 76 | 32.3 | 59 | 25.1 |
| 61–70 | 76 | 32.3 | 37 | 15.7 |
| >70 | 38 | 16.2 | 16 | 6.9 |
| Marital status | | | | |
| Single | 12 | 5.1 | 34 | 14.4 |
| Married | 176 | 74.9 | 160 | 68.1 |
| Divorced | 17 | 7.2 | 23 | 9.8 |
| Widowed | 30 | 12.8 | 18 | 7.7 |
| Educational status | | | | |
| Unable to read and write | 47 | 20.0 | 54 | 23.0 |
| Read and write | 35 | 14.9 | 31 | 13.2 |
| Primary school | 39 | 16.6 | 45 | 19.1 |
| Secondary school | 28 | 11.9 | 22 | 9.4 |
| Preparatory school | 39 | 16.6 | 29 | 12.3 |
| College or university completed | 47 | 20.0 | 54 | 23.0 |
| Occupational status | | | | |
| Government Employee | 49 | 20.9 | 57 | 24.3 |
| Retired | 59 | 25.1 | 34 | 14.4 |
| Housewife | 45 | 19.1 | 43 | 18.3 |
| Daily labourer | 9 | 3.8 | 16 | 6.8 |
| Merchant | 44 | 18.7 | 32 | 13.6 |
| Farmer | 29 | 12.4 | 53 | 22.6 |
| Residence | | | | |
| Urban | 196 | 83.4 | 170 | 72.3 |
| Rural | 39 | 16.6 | 65 | 27.7 |
| Average family income | | | | |
| 500–1000 ETB | 22 | 9.4 | 38 | 16.2 |
| 1001–2000 ETB | 67 | 28.5 | 55 | 23.4 |
| 2001–3000 ETB | 41 | 17.4 | 45 | 19.1 |
| >3000 ETB | 105 | 44.7 | 97 | 41.3 |

ETB = Ethiopian Birr

government employees. On average, 44.7% of the cases and 41.3% of the controls earned more than 3,000 ETB each month (Table 1).

## Behavioural and psychosocial characteristics of the study participants

According to our findings, most cases (about 81%) and controls (almost 80%) were not khat chewers. Hazardous alcohol consumers comprised one-quarter of the cases (25.1%) and almost one-third of the controls (28.5%). Lower medication adherence was reported in slightly more than half of the cases (47.2%) and two-fifths of the controls (40%). Concerning

adherence to a healthy diet, more than half of the cases (54.9%) and over three-fifths of the controls (64.5%) had a good level of adherence to a healthy diet. Almost 80% of the cases and about 85% of the controls did not adhere to SBGM. Additionally, over the past seven days, nearly half of the cases (49.8%) and more than a quarter (28.9%) of controls reported lower physical activity levels. Depression was experienced by (166 [70.6%]) of the cases and (122 [51.9%]) of the controls. Regarding social support, (102 [43.4%]) of the cases and (106 [45.1%]) of the controls had received a moderate level of social support from their family, friends, and neighbours (Table 2).

## Clinical characteristics of the study participants

Regarding the type of DM, 96.6% of the cases and 81.7% of the controls had type-2 diabetes. The median (IQR) age of cases at diagnosis of DM was 52 (44–59), and the mean (± SD) age of the controls at diagnosis of DM was 44.49 ± (11.0). Concerning the type of DM treatment, most cases (83%) and nearly three-fourths of the controls (71.9%) were on oral hypoglycemic agents. In this study, both cases and controls have a similar median (IQR) duration of diabetes since diagnosis, which was eight years (5–12). More than half of the cases (120 [51.1%]) and more than one-quarter of the controls (66 [28.1%]) reported that they had a family history of hypertension.

Concerning education about diabetes, more than half of the cases (134 [57%]) and nearly three-quarters of the controls (170 [72.3%]) had received diabetes education. Regarding glycemic control, about 54% of the cases and 65.1% of the controls had reasonable glycemic control. In this study, two-fifths of the cases (40%) and more than half of the controls (56.2%) had a healthy weight. While over half of the controls (51.5%) and more than three-fifths of the cases (68.5%) had above-normal waist circumference, nearly half of the controls (47.7%) and less than three-fourth of the cases (68.1%) presented with waist to hip ratio values above average (Table 3).

## Determinants of hypertension among patients with DM

In the bivariate logistic regression analysis, participant's age, marital status, educational status, average family income, non-adherence to diabetic medication, lower level of physical activity, non-adherence to a healthy diet, depression, higher BMI, above the average waist circumference, above the standard waist to hip ratio, family history of hypertension, not having diabetic health education, a longer duration of diabetes, presence of comorbidity other than hypertension, presence of diabetes-related complications and poor glycemic control shows a statistically significant association at P-value < 0.2.

After controlling for the potential confounders, age, physical inactivity, depression, family history of hypertension, not having diabetic health education, duration of diabetes, and poor glycemic control were significant factors at a 5% significance level in the final multivariable logistic regression model.

Compared to individuals with diabetes aged below 50 years, those within the age groups 51–60 years, 61–70 years, and above 70 years were 3.33 times [AOR = 3.331, 95% CI (1.92–5.78)], 3.99 times [AOR = 3.99, 95% CI (2.14–7.46)] and 2.95 times [AOR = 2.95, 95% CI (1.25–5.98)] are more likely to have hypertension.

The odds of having hypertension were also 1.82 times [AOR = 1.82, 95% CI (1.00–3.31)] higher among individuals with diabetes who had lower levels of physical activity compared to those with higher levels of physical activity. Additionally, individuals with diabetes who experienced depression were also twice [AOR = 2.00, 95% CI (1.24–3.21)] more likely to be hypertensive than those with diabetes who had no depression.

**Table 2. Psychosocial and behavioural characteristics of study participants.**

| Variables | Cases (n = 235) | | Controls (n = 235) | |
|---|---|---|---|---|
| | **Frequency** | **%** | **Frequency** | **%** |
| Ever chewed khat | | | | |
| Yes | 45 | 19.1 | 48 | 20.4 |
| No | 190 | 80.9 | 187 | 79.62 |
| Frequency of khat chewing | | | | |
| Less than once a month | 2 | 4.4 | 3 | 6.3 |
| One to three times per month | 1 | 2.2 | 8 | 16.7 |
| Once a week | 12 | 26.7 | 16 | 33.3 |
| Two to four times per week | 16 | 35.6 | 13 | 27.1 |
| Daily | 14 | 31.1 | 8 | 16.6 |
| Hazardous alcohol consumption | | | | |
| Yes | 59 | 25.1 | 67 | 28.5 |
| No | 176 | 74.9 | 168 | 71.5 |
| Alcohol dependency | | | | |
| Low Risk | 176 | 74.9 | 168 | 71.5 |
| Medium Risk | 32 | 13.6 | 43 | 18.3 |
| High Risk | 9 | 3.8 | 10 | 4.3 |
| Addiction Likely | 18 | 7.7 | 14 | 6.0 |
| Adherence to diabetic medication | | | | |
| Low | 111 | 47.2 | 94 | 40.0 |
| Medium | 75 | 31.9 | 79 | 33.6 |
| Good | 49 | 20.9 | 62 | 26.4 |
| Adherence to a healthy diet | | | | |
| Good | 151 | 64.3 | 129 | 54.9 |
| Poor | 84 | 35.7 | 106 | 45. |
| Adherence to self-blood glucose monitoring | | | | |
| Adhered | 36 | 15.3 | 48 | 20.4 |
| Did not adhere | 199 | 84.7 | 187 | 79.6 |
| Levels of physical activity | | | | |
| Low | 118 | 50.2 | 68 | 28.9 |
| Moderate | 75 | 31.9 | 100 | 42.6 |
| High | 42 | 17.9 | 67 | 28.5 |
| Presence of depression | | | | |
| Yes | 166 | 70.6 | 122 | 51.9 |
| No | 69 | 29.4 | 113 | 48.1 |
| Severity of depression | | | | |
| No | 69 | 29.4 | 113 | 48.1 |
| Mild | 78 | 33.2 | 64 | 27.2 |
| Moderate | 56 | 23.8 | 31 | 13.2 |
| Moderately severe | 25 | 10.6 | 20 | 8.5 |
| Sever | 7 | 3.0 | 7 | 3.0 |
| Depression-related level of difficulty in doing work | | | | |
| Not difficult at all | 155 | 66.0 | 133 | 56.6 |
| Somewhat difficult | 51 | 21.7 | 74 | 31.5 |
| Very difficult | 22 | 9.3 | 25 | 10.6 |
| Extremely difficult | 7 | 3.0 | 3 | 1.3 |
| Social Support | | | | |

*(Continued)*

**Table 2.** (Continued)

| Variables | Cases (n = 235) | | Controls (n = 235) | |
|---|---|---|---|---|
| | **Frequency** | **%** | **Frequency** | **%** |
| Poor | 79 | 33.6 | 67 | 28.5 |
| Moderate | 102 | 43.4 | 106 | 45.1 |
| Strong | 54 | 23.0 | 62 | 26.4 |

Regarding family history, individuals with diabetes who had a family history of hypertension were 2.13 times more likely [AOR = 2.13, 95% CI (1.34–3.37)] to have hypertension compared to individuals with diabetes who did not have a family history of hypertension. Individuals with diabetes who did not attend diabetic education were 1.87 times [AOR = 1.87, 95% CI (1.18–2.96)] more likely to have hypertension than individuals with diabetes who attend diabetic education.

The odds of having hypertension were also 1.99 times higher among diabetic patients with duration since diagnosis of DM >10 years as compared to diabetic patients whose DM was less than five years [AOR = 1.99, 95% CI (1.05–3.79)]. Regarding glycemic control, diabetic patients with poor glycemic control had a 1.57-fold increased risk of developing hypertension compared to diabetic patients with reasonable glycemic control [AOR = 1.57, 95% CI (1.01–2.45)] (Table 4).

## Discussion

This study's primary purpose was to assess hypertension determinants among individuals with diabetes attending chronic follow-up clinics in the Amhara region's comprehensive specialized hospitals.

Compared to individuals with diabetes under 50 years, those within the age groups 51–60 years, 61–70 years, and above 70 years were more likely to have hypertension. This finding agrees with studies conducted in Malaysia [14], Emirates [18], Republic of Benin [19], and Ethiopia [6, 11, 21, 22], and they stated that older age (age > 50 years) is associated with an increased risk of having hypertension. This finding might be due to age-related changes in the vascular system that occur in older individuals with diabetes that lead to stiffening and thickening of the artery layers, alteration in renal and sodium metabolism, and modifications to the renin-angiotensin-aldosterone system. These modifications eventually predispose to high blood pressure [43, 44]. Moreover, the probable explanation might also be due to easy susceptibility to pathological conditions, other DM-related complications, and non-adherence to the treatments.

Individuals with diabetes who experienced depression were more likely to be hypertensive than individuals with diabetes who did not experience depression. This finding might result from physiological mechanisms that cause prolonged activation of the sympathetic nervous system and the release of catecholamines, leading to insulin resistance and high blood pressure [45]. Besides, depression is negatively associated with poor adherence to DM treatment regimen, self-care aspects, medical appointment attendance, and quality of life.

Concerning glycemic control, individuals with diabetes who had poor glycemic control had a higher risk of developing hypertension than individuals with diabetes who had reasonable glycemic control. This finding agrees with studies conducted in Debretabor [6] and Tigray, Ethiopia [23]. This association might be attributed to persistent hyperglycemia. A high level of glucose traps circulating low-density lipoprotein, which promotes cholesterol deposition in the intima and induces the formation of atheroma on the artery walls and subsequent

**Table 3. Clinical characteristics of study participants.**

| Variables | Cases (n = 235) | | Controls (n = 235) | |
|---|---|---|---|---|
| | Frequency | % | Frequency | % |
| Type of DM | | | | |
| Type 1 | 8 | 3.4 | 43 | 18.3 |
| Type 2 | 227 | 96.6 | 192 | 81.7 |
| Type of DM treatment | | | | |
| Oral medication | 195 | 83.0 | 169 | 71.9 |
| Insulin therapy | 25 | 10.6 | 58 | 24.7 |
| Oral medication plus insulin | 15 | 6.4 | 8 | 3.4 |
| Duration of diabetes | | | | |
| <5year | 46 | 19.5 | 85 | 36.2 |
| 5–10year | 108 | 46.0 | 102 | 43.4 |
| >10year | 81 | 34.5 | 48 | 20.4 |
| Family history of DM | | | | |
| Yes | 105 | 44.7 | 111 | 47.2 |
| No | 130 | 55.3 | 124 | 52.8 |
| Family history of HTN | | | | |
| Yes | 120 | 51.1 | 66 | 28.1 |
| No | 115 | 48.9 | 169 | 71.9 |
| Kept follow-up appointments with a physician | | | | |
| Yes | 206 | 87.7 | 195 | 83.0 |
| No | 29 | 12.3 | 40 | 17.0 |
| Had health education about diabetes | | | | |
| Yes | 134 | 57.0 | 170 | 72.3 |
| No | 101 | 43.0 | 65 | 27.7 |
| Presence of comorbidity other than HTN | | | | |
| Yes | 110 | 46.8 | 55 | 23.4 |
| No | 125 | 53.2 | 180 | 76.6 |
| Specific types of comorbidities * | | | | |
| Cardiovascular disease | 79 | 71.8 | 42 | 76.4 |
| Respiratory disease | 17 | 15.5 | 4 | 7.3 |
| Renal disease | 19 | 17.3 | 11 | 20 |
| Neurological disease | 5 | 7.3 | 8 | 9.1 |
| Presence of Diabetes-related complications | | | | |
| Yes | 57 | 24.3 | 35 | 14.9 |
| No | 178 | 75.7 | 200 | 85.1 |
| Type of diabetes-related complications * | | | | |
| Diabetic retinopathy | 5 | 14.3 | 9 | 15.8 |
| Diabetic nephropathy | 12 | 34.3 | 26 | 45.6 |
| Diabetic neuropathy | 13 | 37.1 | 19 | 33.3 |
| Sexual dysfunction | 7 | 20.0 | 3 | 5.3 |
| Glycemic control | | | | |
| Good | 126 | 53.6 | 153 | 65.1 |
| Poor | 109 | 46.4 | 82 | 34.9 |
| Body mass index | | | | |
| Underweight | 1 | 0.4 | 6 | 2.5 |
| Healthy weight | 94 | 40.0 | 132 | 56.2 |
| Overweight | 105 | 44.7 | 70 | 29.8 |

(*Continued*)

**Table 3.** (Continued)

| Variables | Cases (n = 235) | | Controls (n = 235) | |
|---|---|---|---|---|
| | Frequency | % | Frequency | % |
| Obese | 35 | 14.9 | 27 | 11.5 |
| Waist circumference | | | | |
| Normal | 74 | 31.5 | 114 | 48.5 |
| Above normal | 161 | 68.5 | 121 | 51.5 |
| Waist to Hip Ratio | | | | |
| Normal | 75 | 31.9 | 123 | 52.3 |
| Above Normal | 160 | 68.1 | 112 | 47.7 |

*Indicates that the total will not add up to 235 (100%) for cases and 235 (100%) for controls, as multiple responses were possible in these categories

hypertension [46–48]. Furthermore, an increased blood glucose level increases extracellular fluid's osmolality, causing water to shift from the intracellular to the extracellular space, resulting in volume expansion and high blood pressure.

The odds of hypertension among individuals with diabetes who had lower levels of physical activity were higher than those with higher levels of physical activity. This finding agrees with studies conducted in Benghazi, Libya [16] and Tigray, Ethiopia [23]. This association may have occurred because exercise can normalize BMI, improve glucose tolerance and insulin sensitivity, reduce systemic vascular resistance, and improve insulin sensitivity. Furthermore, exercise can improve lipid metabolism, resulting in better glycemic control and an optimal blood pressure level [49].

Regarding family history, individuals with diabetes who had a positive family history of hypertension were more likely to have hypertension than those who had no family history of hypertension. This finding is consistent with studies conducted in Southern Ethiopia [50] and Jordan [24]. This finding might result from genetic factors associated with high blood pressure in individuals with diabetes, such as high sodium-lithium counter-transport, elevated uric acid levels, high-fasting plasma insulin concentrations, and oxidative stress.

The odds of hypertension among individuals with diabetes who did not attend diabetic education were higher than those with diabetes who attended diabetic education. This finding is consistent with a study conducted in Tigray, Ethiopia [23]. The above finding might have occurred due to diabetic education being a key factor for good blood pressure and glucose management and positively affecting patient health [40, 51].

In this study, individuals with diabetes who had diabetes for above ten years had a higher risk of having hypertension than those who had diabetes for less than five years. This finding is supported by a study conducted in the Republic of Benin [19] and Adama, Ethiopia [17]. This finding might be because as diabetes duration increases, changes caused by diabetes mellitus, such as micro-vascular damage, sympathetic damage, an enhanced renin-angiotensin system, and decreased insulin sensitivity, will aggravate hypertension [46, 52].

Using information from patient's medical records and physical measurements to gather information rather than relying only on self-reported information, as well as being a multicenter study, were significant strengths of this study. To the best of our knowledge, this study is also the first study to assess the effect of depression on causing hypertension among individuals with DM. Since the study was hospital-based, the finding may not be generalized to the general population, and using fasting blood glucose levels instead of HgA1C were limitations of this study. Additionally, the use of p<0.2 as the stopping rule to identify a candidate set of covariates in the bivariate model to be included in the multivariable model might not provide

**Table 4. Bivariate and multivariable logistic regression analysis results.**

| Variables | Cases | Controls | COR | AOR | P |
|---|---|---|---|---|---|
| | (n = 235) | (n = 235) | (95%CI) | (95%CI) | Value |
| Age (years) | | | | | |
| >70 | 38 (16.2) | 16 (6.8) | 6.492(3.30, 12.77) | 2.95 (1.25, 6.92) | 0.013** |
| 61–70 | 76 (32.3) | 37 (15.7) | 5.61 (3.34, 9.45) | 3.99 (2.14, 7.46) | < 0.001** |
| 51–60 | 76 (32.3) | 59 (25.1) | 3.52 (2.17,5.70) | 3.33 (1.92, 5.78) | < 0.001** |
| ≤ 50 | 45 (19.2) | 123 (52.3) | 1 | 1 | |
| Marital status | | | | | |
| Married | 176 (74.9) | 160 (68.1) | 3.11 (1.56, 6.23) | 1.56 (0.67, 3.64) | 0.305 |
| Divorced | 17 (7.2) | 23 (9.8) | 2.09 (0.84, 5.19) | 1.15 (0.39, 3.39) | 0.797 |
| Widowed | 30 (12.8) | 18 (7.7) | 4.72 (1.96, 11.39) | 1.10 (0.37, 3.32) | 0.865 |
| Single | 12 (5.1) | 34 (14.4) | 1 | 1 | |
| Educational status | | | | | |
| Unable to read and write | 47 (20.0) | 54 (23.0) | 1.00 (0.58, 1.74) | 0.89 (0.41, 1.96) | 0.779 |
| Read and write | 35 (14.9) | 31 (13.2) | 1.29 (0.69, 2.4) | 1.05 (0.46, 2.36) | 0.915 |
| Primary school | 39 (16.6) | 45 (19.1) | 0.99 (0.55, 1.77) | 0.96 (0.45, 2.05) | 0.910 |
| secondary school | 28 (11.9) | 22 (9.4) | 1.46 (0.74, 2.89) | 1.16 (0.49, 2.74) | 0.744 |
| preparatory school | 39 (16.6) | 29 (12.3) | 1.54 (0.83, 2.87) | 1.13 (0.51, 2.48) | 0.767 |
| College or university | 47 (20.0) | 54 (23.0) | 1 | 1 | |
| Average family income (ETB) | | | | | |
| >3000 | 105 (44.7) | 97 (41.3) | 1.86 (1.03, 3.38) | 1.609 (0.72, 6.62) | 0.251 |
| 2001–3000 | 41 (17.4) | 45 (19.1) | 1.57 (0.80, 3.08) | 1.264 (0.54, 2.54) | 0.592 |
| 1001–2000 | 67 (28.5) | 55 (23.4) | 2.10 (1.11, 3.96) | 1.77 (0.81, 3.86) | 0.151 |
| 500–1000 | 22 (9.4) | 38 (16.2) | 1 | 1 | |
| Adherence to diabetic medication | | | | | |
| Low | 111 (47.2) | 94 (40) | 1.49 (0.93, 2.37) | 1.15 (0.64, 2.04) | 0.646 |
| Medium | 75 (31.9) | 79 (33.6) | 1.20 (0.73, 1.961) | 0.93 (0.51, 1.69) | 0.802 |
| Good | 49 (20.9) | 62 (26.4) | 1 | 1 | |
| Levels of physical activity | | | | | |
| Low | 117 (49.8) | 68 (28.9) | 2.74 (1.69, 4.47) | 1.82 (1.00, 3.31) | 0.049** |
| Moderate | 76 (32.3) | 100 (42.6) | 1.21 (0.75,1.97) | 1.02 (0.57, 1.84) | 0.947 |
| High | 42 (17.9) | 67 (28.5) | 1 | 1 | |
| Adherence to a healthy diet | | | | | |
| Poor | 84 (35.7) | 106 (45.1) | 0.67 (0.47, 0.98) | 0.75 (0.47, 1.19) | 0.219 |
| Good | 151 (64.3) | 129 (54.9) | 1 | 1 | |
| Depression | | | | | |
| Yes | 166 (70.6) | 122 (51.9) | 2.22 (1.52, 3.26) | 2.00 (1.24, 3.21) | 0.004** |
| No | 69 (29.4) | 113 (48.1) | 1 | 1 | |
| Waist Circumference | | | | | |
| Normal | 74 (31.5) | 114 (48.5) | 1 | 1 | |
| Above normal | 161 (68.5) | 121 (51.5) | 2.04 (1.40, 2.98) | 0.85 (0.37, 1.97) | 0.704 |
| Waist to Hip Ratio | | | | | |
| Normal | 75 (31.9) | 123 (52.3) | 1 | 1 | |
| Above Normal | 160 (68.1) | 112 (47.7) | 2.34 (1.61, 3.41) | 1.84 (0.80, 4.23) | 0.150 |
| Family history of HTN | | | | | |
| Yes | 120 (51.1) | 66 (28.1) | 2.67 (1.82, 3.19) | 2.13 (1.34, 3.37) | ≤ 0.001** |
| No | 115 (48.9) | 169 (71.9) | 1 | 1 | |
| Had health education about diabetes | | | | | |

*(Continued)*

**Table 4.** (Continued)

| Variables | Cases | Controls | COR | AOR | P |
|---|---|---|---|---|---|
| | (n = 235) | (n = 235) | (95%CI) | (95%CI) | Value |
| Yes | 134 (57.0) | 170 (72.3) | 1 | 1 | |
| No | 101 (43.0) | 65 (27.7) | 1.97 (1.34, 2.89) | 1.87 (1.18, 2.96) | 0.008** |
| Duration of DM | | | | | |
| >10year | 81 (34.5) | 48 (20.4) | 3.11 (1.88, 5.17) | 1.99 (1.05, 3.79) | 0.036** |
| 5–10year | 108 (46.0) | 102 (43.4) | 1.95 (1.24, 3.06) | 1.51 (0.88, 2.61) | 0.137 |
| <5year | 46 (19.6) | 85 (36.2) | 1 | 1 | |
| Presence of comorbidity other than hypertension | | | | | |
| Yes | 110 (46.8) | 55 (23.4) | 2.880 (1.94, 4.28) | 1.38 (0.84, 2.27) | 0.206 |
| No | 125 (53.2) | 180 (76.6) | 1 | 1 | |
| Presence of diabetes-related complications | | | | | |
| Yes | 57 (24.3) | 35 (14.9) | 1.82 (1.15, 2.92) | 0.79 (0.45, 1.42) | 0.444 |
| No | 178 (75.7) | 200 (85.1) | 1 | 1 | |
| Glycemic control | | | | | |
| Good | 126 (53.6) | 153 (65.1) | 1 | 1 | |
| Poor | 109 (46.4) | 82 (34.9) | 1.61 (1.11, 2.34) | 1.57 (1.01, 2.45) | 0.046** |

** Indicates that variables are statistically significant at a P value < 0.05

an optimal variable selection for the covariates, although previous studies have provided a strong recommendation for using p-values within the range of 0.15–0.20 [53] as used in our study.

## Conclusion

In this study, determinants that increase the risk of hypertension among people with diabetes mellitus were older age, physical inactivity, depression, family history of hypertension, not having diabetic health education, longer duration of diabetes, and poor glycemic control.

## Supporting information

**S1 Data.**
(SAV)

## Acknowledgments

We are grateful to Bahir Dar University, the College of Medicine and Health Science, and the Department of Adult Health Nursing for letting us conduct this research. We extend our thanks to the study participants for their willingness to participate.

## Author Contributions

**Conceptualization:** Makda Abate Belew, Eleni Dagnaw Abeje.

**Data curation:** Makda Abate Belew, Dawit Algaw Ayele.

**Formal analysis:** Makda Abate Belew.

**Methodology:** Makda Abate Belew, Melsew Dagne Abate.

**Software:** Rediet Akele Getu.

**Visualization:** Berihun Bantie.

**Writing – original draft:** Makda Abate Belew, Sewnet Getaye Workie.

**Writing – review & editing:** Teshager Woldegiorgis Abate, Alemshet Yirga Berhie.

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
