## [Decision Letter · Decision Letter 0]

19 Aug 2022

PONE-D-22-13803Determinants of Hypertension among adult Diabetes Mellitus patients in Amhara Region comprehensive specialized hospitals, Amhara Region, Ethiopia, 2020/21. A multi-center Unmatched Case-Control StudyPLOS ONE

Dear Mr Abate,

Thank you for submitting your manuscript to PLOS ONE. After careful consideration, we feel that it has merit but does not fully meet PLOS ONE’s publication criteria as it currently stands. Therefore, we invite you to submit a revised version of the manuscript that addresses the points raised during the review process.

We look forward to receiving your revised manuscript.

Kind regards,

Jibril Mohammed, BSc, MSc, PhD

Academic Editor

PLOS ONE

Journal Requirements:

2. You indicated that you had ethical approval for your study. In your Methods section we have noted that the interquartile range for age was  (12- 66), please ensure you have also stated whether you obtained consent from parents or guardians of the minors included in the study or whether the research ethics committee or IRB specifically waived the need for their consent.

Whilst you may use any professional scientific editing service of your choice, PLOS has partnered with both American Journal Experts (AJE) and Editage to provide discounted services to PLOS authors. Both organizations have experience helping authors meet PLOS guidelines and can provide language editing, translation, manuscript formatting, and figure formatting to ensure your manuscript meets our submission guidelines. To take advantage of our partnership with AJE, visit the AJE website (http://aje.com/go/plos) for a 15% discount off AJE services. To take advantage of our partnership with Editage, visit the Editage website (www.editage.com) and enter referral code PLOSEDIT for a 15% discount off Editage services. If the PLOS editorial team finds any language issues in text that either AJE or Editage has edited, the service provider will re-edit the text for free.

Additional Editor Comments:

Dear Authors: Please find the comments received from two reviewers. In general, your paper appears to be lacking in so many areas and a lot of work has to be done for me to reconsider the possibility of accepting it for publication. I also agree with the reviewers that the analyses lacks clarity and there is a need for substantial improvement in writing style of the manuscript (English language editing!).

Reviewers' comments:

Reviewer's Responses to Questions

**Comments to the Author**

1. Is the manuscript technically sound, and do the data support the conclusions?

Reviewer #1: Partly

Reviewer #2: Yes

2. Has the statistical analysis been performed appropriately and rigorously? 

Reviewer #1: Yes

Reviewer #2: I Don't Know

3. Have the authors made all data underlying the findings in their manuscript fully available?

Reviewer #1: Yes

Reviewer #2: Yes

4. Is the manuscript presented in an intelligible fashion and written in standard English?

Reviewer #1: No

Reviewer #2: Yes

5. Review Comments to the Author

Reviewer #1: The paper was meant to add an important piece of information to the literature on diabetes complicated by high blood pressure among persons with diabetes from a region in an African country, but the justification for the study is rather incomplete. In addition, there are lots of inconsistencies with the use of terms/nomenclatures both in the text and in the tables and the discussion is POORLY written. The authors MUST ponder the following:

1. In the introduction, there is need to include what has been documented about the determinants of HTN among persons with DM. The authors attempted to capture these in paragraph 4 of the introduction but more cogent information on these determinants must be included.

2. In the introduction, authors MUST explain the social context in the previous (Tigray study) study and what makes the social context in their study broader than the one in the Tigray study? What makes the Amhara region more diverse than Tigray? What makes the Amhara region of broader social context than Tigray?

3. Regarding justification of the study, of what significance is making such data (determinants of HTN among diabetes patients) available? Whom does it benefit and in what way?

3. Did authors consider HTN existing at the time of or after the diagnosis of DM?

4. Authors must be consistent with the use of terms/nomenclature. E.g., “individuals or persons with diabetes” in lieu of “diabetic individuals or clients”

5. There are too many long-winded statements that authors MUST reorganize them. In addition, the manuscript MUST be submitted for English language editing.

6. The discussion MUST undergo a major overhaul if the manuscript is to be considered for publication.

Authors MUST improve the quality of the manuscript by addressing all the issues raised (see my comments in the manuscript) and resubmit it.

Reviewer #2: The introduction is very broad and as the research question is searching for determinants of hypertension it should be written more focused on already known determinants and what will be new from this study.

Methods/results:

Exclusion criteria: those who have a history of hypertension at the time of diagnosis of DM: why is this an exclusion?

Some operational definitions need more explanation (eg physical activity and diet regimen)

Statistical analysis: why cut off of 0.2 voor regression (in most cases 0.15)?

IN the results the logistic regression is mentioned and than there is one with correction for potential confounders (most of the entered parameters in the regression). Can you explain a little bit better how the statistics is performed and how the results are expressed.

The results are not clearly written. It is not always very clear to the reader how the results are shown and what this means. This should be rewritten and made more focussed.

The discussion: This should start with the main findings. Than you can discuss the main points, but discuss the new elements. If you compare with other research you will find that duration of diabetes is a predictor in many countries. This is not very new. Perhaps the weigth of this factor in the Nigerian population is different from other countries. This can be more discussed in depth.

Figures and tables: nice, but sometimes you need to give more information in the legend.

References: OK

6. PLOS authors have the option to publish the peer review history of their article (what does this mean?). If published, this will include your full peer review and any attached files.

Reviewer #1: No

Reviewer #2: **Yes: **Patrick Calders

---

## [Author Response · Author response to Decision Letter 0]

27 Oct 2022

Author’s response to editor and reviewers 

Dear editor and reviewers, 

We appreciate your review and the significant points you made in the paper.

We read the reviewer's comments carefully and we are hoping that the newly submitted revised version will be viewed as being better than the original. The authors very much appreciate the reviewer's suggestions and comments. These comments significantly raised the quality of the manuscript. 

Please find the responses to the issues raised under each of the points below.

The author's responses to the comments are in italics below, and the necessary changes in the main manuscript are highlighted in color. In addition, we made a few other changes to the paper to improve our readers' understanding, which were also highlighted in red.

PONE-D-22-13803

Determinants of Hypertension among adult Diabetes mellitus patients in Amhara Region comprehensive specialized hospitals, Amhara Region, Ethiopia, 2020/21. A multi-centre Unmatched Case-Control Study.

PLOS ONE

Dear Mr Abate,

Thank you for submitting your manuscript to PLOS ONE. After careful consideration, we feel that it has merit but does not fully meet PLOS ONE's publication criteria as it currently stands. Therefore, we invite you to submit a revised version of the manuscript that addresses the points raised during the review process.

Included 

Included 

Included 

We look forward to receiving your revised manuscript.

Kind regards, 

Jibril Mohammed, BSc, MSc, PhD

Academic Editor

PLOS ONE

Journal requirements 

Response: we accepted your suggestion and revised the manuscript following PloS One's style requirements. 

2. You indicated that you had ethical approval for your study. In your Methods section we have noted that the interquartile range for age was (12-66), please ensure you have also stated whether you obtained consent from parents or guardians of the minors included in the study or whether the research ethics committee or IRB specifically waived the need for their consent.

Response: thank you for bringing this to our attention, and please accept our apologies for making such a mistake while writing the result. The actual interquartile range for age was (52 -66=14).

Whilst you may use any professional scientific editing service of your choice, PLOS has partnered with both American Journal Experts (AJE) and Editage to provide discounted services to PLOS authors. Both organizations have experience helping authors meet PLOS guidelines and can provide language editing, translation, manuscript formatting, and figure formatting to ensure your manuscript meets our submission guidelines. To take advantage of our partnership with AJE, visit the AJE website (http://aje.com/go/plos) for a 15% discount off AJE services. To take advantage of our partnership with Editage, visit the Editage website (www.editage.com) and enter referral code PLOSEDIT for a 15% discount off Editage services. If the PLOS editorial team finds any language issues in text that either AJE or Editage has edited, the service provider will re-edit the text for free.

Response: Thank you for suggesting that the English writing in this manuscript be revised. An expert editor, Mr Abera Lambebo, a PhD scholar in Human Nutrition and an assistant professor of public health nutrition at Debre Berhan University, edited the revised manuscript. Use this email address to get in touch with him if you need more details: lambebo70@gmail.com. In addition, we used "Grammarly" online software to edit the spelling, grammar, and language usage. Accordingly, we have got a 93% quality score in the draft manuscript. Then taking the comment raised by online Grammarly software, we addressed most of the issues, left some that were illogical. Later, we got a 99% quality score in the final revised draft.

Response: The datasets used and analyzed during the current study are not publicly available because the hospitals and participants did not ethically permit the researchers to disclose the data publicly. However, the dataset is available from the corresponding author if future researchers have a reasonable justification. Individual researchers granted permission to use the data are not allowed to make the data (in any form) available to third parties. Any third party interested in the data must contact the corresponding author directly.

Additional Editor Comments: 

Comments to the Author

1. Is the manuscript technically sound, and do the data support the conclusions?

Reviewer #1: Partly 

Reviewer #2: Yes________________________________________

2. Has the statistical analysis been performed appropriately and rigorously?

Reviewer #1: Yes 

Reviewer #2: I Don't Know

3. Have the authors made all data underlying the findings in their manuscript fully available?

Reviewer #1: Yes 

Reviewer #2: Yes

4. Is the manuscript presented in an intelligible fashion and written in Standard English?

Reviewer #1: No

Reviewer #2: Yes

Reviewer #1: The paper was meant to add an important piece of information to the literature on diabetes complicated by high blood pressure among persons with diabetes from a region in an African country, but the justification for the study is rather incomplete. In addition, there are lots of inconsistencies with the use of terms/nomenclatures both in the text and in the tables and the discussion is POORLY written. 

The authors MUST ponder the following:

1. In the introduction, there is need to include what has been documented about the determinants of HTN among persons with DM. The authors attempted to capture these in paragraph 4 of the introduction but more cogent information on these determinants must be included.

Response: Despite implementing different strategies (Behavioral risk reduction and lifestyle modification), many studies reported a high burden of hypertension among people with DM (1-6). So far, some determinants of hypertension in people with diabetes have been identified in some countries (United Arab Emirates, Jordan, Nigeria, Malaysia, Republic of Benin, Libya, Ethiopia), namely, advanced age, low educational status, unhealthy diet, family history of hypertension, poor glycemic control, and duration of diabetes (2-5, 7-9). The most common determinant was a longer duration of diabetes. We aimed to fill the existing research gap in our study area by examining multiple risk factors (depression, harmful alcohol consumption, poor social support, raised BMI, increased waist circumference, and raised waist-to-hip ratio) contributing to the development of hypertension in individuals with DM.

2. In the introduction, authors MUST explain the social context in the previous (Tigray study) study and what makes the social context in their study broader than the one in the Tigray study? What makes the Amhara region more diverse than Tigray? What makes the Amhara region of broader social context than Tigray?

Response: In the previous study conducted in Tigray, they recommend researching the determinants of hypertension among diabetic patients in a broader social context and larger sample size. So, the current study was conducted using a relatively larger sample size and a more diverse social context. Even though the community in Tigray is a lot similar to the Amhara region in terms of sociodemographic and cultural contexts, the study conducted in Tigray included only the central zone (one of the seven Zones in the region). There are only four public hospitals in the zone, namely Aksum University referral hospital, St. Marry General Hospital, Adwa General hospital, and Abyiadi General Hospital, which are found in Axum city, Adwa and Abyiadi towns (10). So, it is difficult to generalize the study's findings to the whole Tigray region since it is not representative of the many hospitals in the region.On the contrary, we believe that the case of the current study includes samples representative of the whole Amhara region (the rural and urban population). The other reason is that the current study is conducted in tertiary level health care specialized hospitals, which serve 3.5-5.0 million populations each, while the previous study was conducted in secondary level general and referral hospitals, which serves 1-1.5 million peoples each (11). So, the researchers believe the current study covers a more diverse and broader social context than the previous study conducted in the central Zone of Tigray, Ethiopia.

3. Regarding justification of the study, of what significance is making such data (determinants of HTN among diabetes patients) available? Whom does it benefit and in what way?

Response: One of the International Diabetes Federation’s suggested actions to lower the risk of CVD outcomes and chronic kidney disease among individuals with DM is to prevent the development of HTN. Although the need to reduce HTN burden among DM patients is extensively studied, how we reduce it in this population calls for further evidence. Despite the disproportionately high burden of HTN among individuals with DM, to the best of our knowledge, there are a limited number of studies on determinants of Hypertension in Ethiopia, including our study area (Amhara region) conducted on this specific population. These studies have significant methodological limitations, including flawed participant population selection to explore determinant factors for developing HTN among DM patients. Therefore, this study addressed the issues mentioned above and identified determinants of HTN among DM patients that can be used for the effective prevention and control of the condition. 

3. Did authors consider HTN existing at the time of or after the diagnosis of DM?

Response: Authors consider hypertension after the diagnosis of DM. 

4. Authors must be consistent with the use of terms/nomenclature. E.g., “individuals or persons with diabetes” in lieu of “diabetic individuals or clients”

Response: thank you for your suggestion; We made modifications per the reviewer's comment.

5. There are too many long-winded statements that authors MUST reorganize them. In addition, the manuscript MUST be submitted for English language editing.

Response: thank you for your suggestion, and to the best of our effort, attempts have been made extensively to summarize the concepts. Furthermore, the manuscript is edited using online Grammarly software and an expert editor. 

6. The discussion MUST undergo a major overhaul if the manuscript is to be considered for publication.

Response: comments were accepted, and we made amendments to the main manuscript 

Reviewer #2: 

1. The introduction is very broad and as the research question is searching for determinants of hypertension it should be written more focused on already known determinants and what will be new from this study.

Response: comment was accepted, and an amendment was made as per the reviewers comment.

2. Methods/results:

Exclusion criteria: those who have a history of hypertension at the time of diagnosis of DM: why is this exclusion?

Response: We excluded those with a history of hypertension during their DM diagnosis from our study. Because our study is looking for determinants of hypertension among people with diabetes, these individuals already develop HTN prior to diabetes. So, we excluded them to avoid selection bias, which occurs when there is a systemic error in determining cases or controls in case-control studies (ascertainment bias).

Some operational definitions need more explanation (e.g. physical activity and diet regimen)

Response: we accept your comment, and adherence to physical activity and adherence to a healthy diet were operationalized as follows;

Physical activity: In our study, we have used a short form of international physical activity questionnaire to assess participants' level of physical activity. In accordance with the international physical activity questionnaire (IPAQ) screening protocol, we operationalize adherence to a higher, moderate and lower level of physical activity as follow:

1. Adherence to a higher level of physical activity: - 

Individuals with diabetes were considered to have a higher level of physical activity if they 

• Have done vigorous physical activity for at least three days and achieve a minimum total physical activity of at least 1500 metabolic equivalent of task (MET) minutes per week. OR 

• Having seven or more days of any combination of walking, moderate intensity or vigorous intensity activities achieving a minimum total physical activity of at least 3000MET minutes in a week (12). 

2. Adherence to a Moderate level of physical activity: - 

Individuals with diabetes were considered to have moderate adherence to physical activity if they

• Engaging in 3 or more days of vigorous-intensity activity and/or walking for at least 30 minutes per day OR 

• Engage in five or more days of moderate-intensity activity and/or walking for at least 30 minutes per day OR 

• Having five or more days of any combination of walking, moderate-intensity or vigorous-intensity activities, achieving a minimum total physical activity of at least 600MET minutes a week (12). 

3. Lower adherence to physical activity: 

• Individuals with diabetes were considered to have lower adherence to physical activity if they were not meeting any of the criteria for either moderate or high levels of physical activity (12).

Dietary adherence 

We used the perceived dietary adherence questionnaire (PDAQ), a nine-item tool to assess adherence to a healthy diet (13). The response is based on a seven-point Likert scale to answer the question, "On how many of the last seven days did you….?" Higher scores reflect higher adherence except for items 4 and 9, which reflect unhealthy choices (foods high in sugar or fat). For these items, higher scores reflect lower adherence. Therefore, for computing a total PDAQ score, the scores for these items were inverted. So, 

• Participants were considered to have good adherence to a healthy diet if they ate a healthy diet for at least four days a week and have poor adherence to a healthy diet if they had eaten a healthy diet for less than four days a week (13).

Statistical analysis: why cut off 0.2 for regression (in most cases 0.15)?

Response: Thank you for pointing this out. Our decision to use a p-value threshold of 0.2 is based on a review of the literature on traditional stopping rules and suggested optimal p-values. The literature on this topic strongly suggests using a p-value in the range of 0.15-0.20 (14), though using a higher significance level has the disadvantage of including some unimportant variables in the model (14). Typically, the traditional stopping rule for the significance level is considered to be between 0.05 and 0.10, but it has also been established that the optimum value of the significance level to decide which variable to include in the multiple regression model is suggested to be p<1(15),which actually exceeds the traditional choices.

The above discussion in the literature is what informed our choice and that is what the authors meant by the statement "we applied the statistical methodology for variable selection using p<0.2 in the bivariate analysis as a threshold to identify a candidate set of variables that will enter the multivariable model which is a standard approach". Our choice of p-value<0.2 also lies between the maximum for the traditional rule and that of suggested p-value<1 which is an acceptable approach.

We revised the manuscript to emphasize the above mentioned issue under the limitations in lines 366 – 370. We added the following sentence: "The use of p<0.2 as the stopping rule to identify candidate set of covariates in the bivariate model to be included in the multivariable model may not provide optimal variable selection for the covariates, despite previous studies providing a strong recommendation for using p-values in the range of 0.15-0.20 as used in our study.”

In the results, the logistic regression is mentioned and then there is one with correction for potential confounders (most of the entered parameters in the regression). Can you explain a little bit better how the statistics is performed and how the results are expressed. 

Response: Confounders are those variables statistically associated with the exposure, cause the outcome of interest, variables that are not on the causal pathway, and variables that could affect the association of the exposure and the outcome variables (16). There are different methods to control confounders at the design stage (randomization, restriction, or matching) and analysis stage (multivariable analysis) (17). In the current study, initially, we have included variables with a p-value less than or equal to 0.2 in the bivariate logistic regression into a multivariable binary logistic regression. Then, we declared significant statistical association at a p-value less than or equal to 0.05 within a 95% confidence interval and interpreted the findings using adjusted odd ratios. So, we tried to control possible confounders in the study during the statistical analysis by using multivariable logistic regression and adjusted odd ratios to interpret and report the results. 

3. The results are not clearly written. It is not always very clear to the reader how the results are shown and what this means. This should be rewritten and made more focused.

Response: comment accepted and amendment was made in the main manuscript as per the reviewer's comment.

4. The discussion: This should start with the main findings. Than you can discuss the main points, but discuss the new elements. If you compare with other research you will find that duration of diabetes is a predictor in many countries. This is not very new. Perhaps the weight of this factor in the Nigerian population is different from other countries. This can be more discussed in depth.

Response: comment was accepted, and an amendment was made in the main manuscript as per the reviewer's comment

5. Figures and tables: nice, but sometimes you need to give more information in the legend.

References: OK

6. PLOS authors have the option to publish the peer review history of their article (what does this mean?). If published, this will include your full peer review and any attached files.

Do you want your identity to be public for this peer review? For information about this choice, including consent withdrawal, please see our Privacy Policy.

Reviewer #1: No 

Reviewer #2: Yes: Patrick Calders

References 

1. Alqudah B, Mahmoud H, Alhusamia S, Sh A, Al L, Alawneh Z. Prevalence of hypertension among diabetic type 2 patients attending medical clinic at prince Hashem Bin. Indian J Med Res Pharm Sci. 2017;4(6):47-54.

2. Abougalambou SSI, Abougalambou AS. A study evaluating prevalence of hypertension and risk factors affecting on blood pressure control among type 2 diabetes patients attending teaching hospital in Malaysia. Diabetes & Metabolic Syndrome: Clinical Research & Reviews. 2013;7(2):83-6.

3. Chinedu A, Nicholas A. Hypertension prevalence and body mass index correlates among patients with diabetes mellitus in Oghara, Nigeria. The Nigerian Journal of General Practice. 2015;13(1):12.

4. Nouh F, Omar M, Younis M. Prevalence of hypertension among diabetic patients in Benghazi: a study of associated factors. Asian Journal of Medicine and Health. 2017:1-11.

5. Dedefo A, Galgalo A, Jarso G, Mohammed A. Prevalence of hypertension and its management pattern among type 2 diabetic patients attending, Adama Hospital Medical College, Adama. J Diabetes Metab. 2018;9(10):1-8.

6. Akalu Y, Belsti Y. Hypertension and Its Associated Factors Among Type 2 Diabetes Mellitus Patients at Debre Tabor General Hospital, Northwest Ethiopia. Diabetes, Metabolic Syndrome and Obesity: Targets and Therapy. 2020;13:1621.

7. Mussa BM, Abduallah Y, Abusnana S. Prevalence of hypertension and obesity among emirati patients with type 2 diabetes. J Diabetes Metab. 2016;7(1):1-5.

8. Amoussou-Guenou D, Wanvoegbe A, Agbodandé A, Dansou A, Tchabi Y, Eyissè Y, et al. Prevalence and risk factors of hypertension in type 2 diabetics in Benin. Journal of Diabetes mellitus. 2015;5(04):227.

9. Mubarak FM, Froelicher ES, Jaddou HY, Ajlouni KM. Hypertension among 1000 patients with type 2 diabetes attending a national diabetes center in Jordan. Annals of Saudi medicine. 2008;28(5):346-51.

10. Tasew H, Zemicheal M, Teklay G, Mariye T. Risk factors of stillbirth among mothers delivered in public hospitals of Central Zone, Tigray, Ethiopia. African health sciences. 2019;19(2):1930-7.

11. World Health O, Alliance for Health P, Systems R. Primary health care systems (primasys): case study from Ethiopia: abridged version. Geneva: World Health Organization, 2017 2017. Report No.: Contract No.: WHO/HIS/HSR/17.8.

12. Forde C. Scoring the international physical activity questionnaire (IPAQ). University of Dublin. 2018.

13. Asaad G, Sadegian M, Lau R, Xu Y, Soria-Contreras DC, Bell RC, et al. The Reliability and Validity of the Perceived Dietary Adherence Questionnaire for People with Type 2 Diabetes. Nutrients. 2015;7(7):5484-96.

14. Hosmer Jr DW, Lemeshow S, Sturdivant RX. Applied logistic regression: John Wiley & Sons; 2013.

15. Chowdhury MZI, Turin TC. Variable selection strategies and its importance in clinical prediction modelling. Family medicine and community health. 2020;8(1).

16. Pourhoseingholi MA, Baghestani AR, Vahedi M. How to control confounding effects by statistical analysis. Gastroenterology and hepatology from bed to bench. 2012;5(2):79-83.

17. Jager KJ, Zoccali C, MacLeod A, Dekker FW. Confounding: What it is and how to deal with it. Kidney International. 2008;73(3):256-60.

---

## [Editor Report · Decision Letter 1]

5 Dec 2022

Determinants of hypertension among diabetes patients attending selected comprehensive specialized hospitals of the Amhara Region, Ethiopia: An unmatched case-control study

PONE-D-22-13803R1

Dear Dr. Abate,

We’re pleased to inform you that your manuscript has been judged scientifically suitable for publication and will be formally accepted for publication once it meets all outstanding technical requirements.

Kind regards,

Jibril Mohammed, BSc, MSc, PhD

Academic Editor

PLOS ONE
---

## [Editor Report · Acceptance letter]

8 Dec 2022

PONE-D-22-13803R1 

Determinants of hypertension among diabetes patients attending selected comprehensive specialized hospitals of the Amhara Region, Ethiopia: An unmatched case-control study 

Dear Dr. Belew:

I'm pleased to inform you that your manuscript has been deemed suitable for publication in PLOS ONE. Congratulations! Your manuscript is now with our production department. 

Kind regards, 

on behalf of

Dr. Jibril Mohammed 

Academic Editor

PLOS ONE